# Selective Blockade of Two Aquaporin Channels, AQP3 and AQP9, Impairs Human Leukocyte Migration

**DOI:** 10.3390/cells14120880

**Published:** 2025-06-11

**Authors:** Sabino Garra, Charlotte Mejlstrup Hymøller, Daria Di Molfetta, Nicola Zagaria, Patrizia Gena, Rosa Angela Cardone, Michael Rützler, Svend Birkelund, Giuseppe Calamita

**Affiliations:** 1Department of Biosciences, Biotechnologies and Environment, University of Bari Aldo Moro, 70125 Bari, Italy; sabino.garra@uniba.it (S.G.); daria.dimolfetta@uniba.it (D.D.M.); nicola.zagaria@uniba.it (N.Z.); annapatrizia.gena@uniba.it (P.G.); rosaangela.cardone@uniba.it (R.A.C.); 2Department of Health Science and Technology, Aalborg University, 9220 Aalborg, Denmark; charlottemh@hst.aau.dk (C.M.H.); sbirkelund@hst.aau.dk (S.B.); 3Apoglyx AB, Medicon Village, 223 81 Lund, Sweden; michael.rutzler@apoglyx.com; 4Division of Biochemistry and Structural Biology, Department of Chemistry, Lund University, 221 00 Lund, Sweden

**Keywords:** host–bacteria relationship, innate immunity, cell motility, white blood cells, phagocytosis, aquaporin inhibitors, LPS, inflammation

## Abstract

Peripheral blood leukocytes are able to migrate to the inflamed tissue, and to engulf and kill invading microbes. This requires rapid modifications of cell morphology and volume through fast movements of osmotic water into or out of the cell. In this process, membrane water channels, aquaporins (AQPs), are critical for cell shape changes as AQP-mediated water movement indirectly affects the cell cytoskeleton and, thereby, the signaling cascades. Recent studies have shown that the deletion or gating of two immune cell AQPs, AQP3 and AQP9, impairs inflammation and improves survival in microbial sepsis. Here, we assessed the expression and distribution of AQP3 and AQP9 in human leukocytes and investigated their involvement in the phagocytosis and killing of the Gram-negative pathogenic bacterium *Klebsiella pneumoniae*, and their role in lipopolysaccharide (LPS)-induced cell migration. By RT-qPCR, AQP3 mRNA was found in peripheral blood mononuclear cells (PBMCs) but it was undetectable in polymorphonuclear white blood cells (PMNs). AQP9 was found both in PBMCs and PMNs, particularly in neutrophil granulocytes. Immunofluorescence confirmed the AQP3 expression in monocytes and, to a lesser degree, in lymphocytes. AQP9 was expressed both in PBMCs and neutrophils. Specific inhibitors of AQP3 (DFP00173) and AQP9 (HTS13286 and RG100204) were used for bacterial phagocytosis and killing studies. No apparent involvement of individually blocked AQP3 or AQP9 was observed in the phagocytosis of *K. pneumoniae* by neutrophils or monocytes after 10, 30, or 60 min of bacterial infection. A significant impairment in the phagocytic capacity of monocytes but not neutrophils was observed only when both AQPs were inhibited simultaneously and when the infection lasted for 60 min. No impairment in bacterial clearance was found when AQP3 and AQP9 were individually or simultaneously blocked. PBMC migration was significantly impaired after exposure to the AQP9 blocker RG100204 in the presence or absence of LPS. The AQP3 inhibitor DFP00173 reduced PBMC migration only under LPS exposure. Neutrophil migration was considerably reduced in the presence of RG100204 regardless of whether there was an LPS challenge or not. Taken together, these results indicate critical but distinct involvements for AQP3 and AQP9 in leukocyte motility, while no roles are played in bacterial killing. Further studies are needed in order to understand the precise ways in which these two AQPs intervene during bacterial infections.

## 1. Introduction

Peripheral blood leukocytes are stimulated by chemoattractants to migrate directionally to the inflamed tissue where they ingest and inactivate potentially harmful microorganisms [1]. This requires a rapid and profound modification of cell shape, volume, and adhesiveness. F-actin plays a pivotal role by responding to and promoting the molecular signals implicated in the structure–function dynamics of the cytoskeleton. Actin polymerization/depolymerization and transmembrane ionic fluxes are believed to lead to rapid changes in osmolality at the cytoplasm adjacent to the leading edge of moving cells. It has been hypothesized that a fast influx of water generates membrane protrusions and an F-actin gradient, followed by G-actin repolymerization for stabilizing the membrane extensions, thereby leading to lamellipodia formation [2]. This hypothesis is supported by the presence of aquaporin water channels at the membrane of the leading edge and impaired migration resulting from their gating or depletion [2,3,4,5,6,7].

Aquaporins (AQPs) are a family of membrane channel proteins found in all areas of the body where they play important roles both in health and disease [8,9,10]. Humans possess 13 distinct AQPs (AQP0-12) grossly subdivided into three groups depending on their molecular selectivity and sequence similarity: (i) the orthodox aquaporins (AQP0, AQP1, AQP2, AQP4, AQP6, and AQP8), (ii) the aquaglyceroporins (AQP3, AQP7, AQP9, and AQP10), and (iii) the superaquaporins (unorthodox aquaporins, AQP11 and AQP12). Furthermore, based on the permeability to hydrogen peroxide and ammonia, classification as peroxiporins (AQP1, AQP3, AQP5, AQP8, AQP9, and AQP11) or ammoniaporins (AQP1, AQP3, AQP6, AQP7, AQP8, and AQP9) has also been used occasionally.

AQPs are reported to play several functions both in adaptive and in innate immunity [11], including immune cell energy production and redox balance [12,13,14], memory CD8+ T-cell longevity [13], bacterial phagocytosis [5], antigen uptake [15], cell maturation [12,16], NLRP3 inflammasome activation [7,17,18,19], and proinflammatory cytokine release [16,17,20]. Important involvements for AQPs have also been described in disorders of the immune system and inflammatory diseases such as sepsis [14,18,20,21,22], systemic inflammatory response syndrome (SIRS) [23], LPS-induced endotoxemia [24], psoriasis [25], hapten-induced contact hypersensitivity [26], asthma [27], severe acute pancreatitis [7], acute respiratory distress syndrome (ARDS) [7,28], neuromyelitis optica spectrum disorder [29,30], and Crohn’s disease [19]. Immune AQPs have been attracting increasing translational interest for several years now, as immune disease markers and drug targets [14,31]. A fair amount of work has already been conducted to define the involvement and modulation of the various AQPs in the immune system. However, molecular mechanisms and downstream signaling pathways in which they are implicated in immune cells are not yet fully known, or there is no general consensus [14]. Furthermore, given their broad selectivity, their precise roles in the immune system, either as water, glycerol, hydrogen peroxide, or other solute channels, or purely as structural scaffolds, remain open questions.

We recently developed and characterized the potent and selective inhibitors of AQP3 [32] and AQP9 [18,33,34,35,36]. With the help of these inhibitors, and by using *Aqp9*^-/-^ knockout mice, we showed the important roles for AQP9 in murine models of endotoxemia [24] and polymicrobial sepsis and systemic inflammation [18]. In the current work, we have assessed the expression of AQP3 and AQP9 in neutrophils and peripheral blood mononuclear cells (PBMCs) isolated from healthy human donors. We studied their involvement in LPS-induced cell migration, and the phagocytosis and killing of the opportunistic Gram-negative pathogen *Klebsiella pneumoniae*. We have observed differences between these two AQP isoforms in cellular expression, and found a significant, but differing involvement in leukocyte migration. Neither AQP appeared to have major functions in the phagocytosis and killing of infecting bacteria.

## 2. Materials and Methods

### 2.1. Aquaporin Inhibitors

HTS13286 and DFP00173 were purchased from Molport (Riga, Latvia). The synthesis and identification of RG100204 have been described previously [18,37].

### 2.2. Blood Collection and Serum Isolation

Whole peripheral blood samples were collected from healthy adult male volunteers. For white blood cells (WBCs), isolation blood was collected by venipuncture into Vacutainer K3-EDTA tubes (Becton, Dickinson and Company, Franklin Lakes, NJ, USA) and processed within 1 h from harvesting. For serum isolation, blood was collected in S-Monovette serum tubes (Sarstedt AG & Co., Nümbrecht, Germany) and incubated for 30 min at room temperature. After centrifugation (2000× *g*, 10 min, at RT), serum was collected. The collection and use of human whole peripheral blood, serum, and plasma were approved by the Regional Ethics Committee of North Denmark region (N-20150073, N-20220006).

### 2.3. Polymorphonuclear, Neutrophil Granulocyte, and PBMC Isolation

WBC fractions were isolated from whole blood samples anticoagulated with K3-EDTA. PMNs were isolated by dextran sedimentation followed by gradient separation. Briefly, blood was mixed 10:1 with Dextran 500 (8% weight/volume of saline solution; Merck Group, Darmstadt, Germany), and, after 30 min, the supernatant made of leucocyte-rich plasma was layered over 3 mL of Lymphoprep (STEMCELL Technologies, Vancouver, BC, Canada) and then centrifuged at 200× *g* for 30 min at RT (low acceleration, with the brake off). The pellet was resuspended in 6 mL of red blood cell (RBC) lysis buffer (0.15 M NH_4_Cl, 0.01 M NaHCO_3_, and 0.0001 M EDTA), and, after 10 min, centrifuged at 300× *g* for 5 min at room temperature (RT). The enriched PMN pellet was resuspended in RPMI 1640, with penicillin (100 U/mL) and streptomycin (100 μg/mL; RPMI1640-Pen-Strept; Euroclone, Pero, Italy). Neutrophil granulocytes were isolated by immunomagnetic negative selection with the EasySep™ Direct Human Neutrophil Isolation Kit (STEMCELL Technologies) according to manufacturer’s instructions. PBMCs were isolated by density gradient centrifugation with SepMate™ tubes (STEMCELL Technologies). Briefly, lymphosep density gradient medium (Biowest, Riverside, MO, USA) was pipetted into a SepMate™ tube through the insert hole, and whole blood diluted 1:1 with phosphate-buffered saline (PBS) (VWR international, Radnor, PA, USA) plus 2% FBS (VWR international, Radnor, PA, USA) was gently pipetted down the side of the tube before spinning at 1200× *g* for 10 min at RT. The upper layer containing plasma and PBMCs was poured into a separate 15 mL tube, before washing once (500× *g*, 10 min, RT) with PBS-2% FBS, and resuspending the pelleted PBMCs in the same medium. Aliquots of each isolated WBC fraction were analyzed by cell count and cell integrity evaluation by a LUNA-II™ counter (Aligned Genetics, Anyang-Si, Republic of Korea).

### 2.4. Total RNA Extraction and RT-qPCR Analysis

For each WBC fraction, total RNA was extracted from 1 × 10^6^ cells using the RNeasy Plus Mini Kit (QIAGEN, Hilden, Germany) according to the manufacturer’s protocol. Total RNA samples were retrotranscribed into cDNA using the High-Capacity cDNA Reverse Transcription Kit (Applied Biosystems, Waltham, MA, USA) as described by the manufacturer. qPCR reactions were carried out using the PowerUp™ SYBR™ Green Master Mix (Applied Biosystems) and specific primers for *AQP1, AQP3*, *AQP5*, *AQP9,* and *β-Actin* (Appendix A). cDNA samples were plated in duplicates and amplified in a QuantStudio™ 1 Real-Time PCR System (Applied Biosystems) with the following program: UDG activation, 50 °C for 2 min; activation, 95 °C for 2 min; and 40 cycles of denaturation (15 s), annealing (15 s), and extension (30 s). AQP3 and AQP9 expression levels were calculated against those of β-actin, used as reference gene according to the 2^−ΔCt^ method.

### 2.5. Immunofluorescence

Isolated PBMCs or neutrophil granulocytes were seeded on SuperFrost Plus™ glass slide (Epredia™, Portsmouth, NH, USA) and incubated at 37 °C for 10 min in humidified atmosphere, then fixed with 4% paraformaldehyde (DDK Italia, Vigevano, Italy) for 20 min before permeabilization with 0.1% Triton X-100 (Merck Group) in PBS for 20 min at RT. Unspecific sites were saturated with 1% BSA (Euroclone) in PBS for 20 min before incubating (90 min at RT) with the rabbit anti-AQP3 (Thermo Fisher Scientific, Waltham, MA, USA) or anti-AQP9 (Merck Group) primary antibodies at a final dilution of 1:60 and 1:100, respectively, in PBS-1% BSA. The antibodies were removed and cells incubated with a 1:1000 dilution of an Alexa Fluor 488 donkey anti-rabbit IgG (Thermo Fisher Scientific) in PBS-1% BSA for 60 min at RT. The negative control condition was made by omitting the primary antibody. Slides were mounted and nuclei stained with Fluoroshield-containing DAPI (Merck Group). Slides were observed with a Leica DMRXA fluorescent microscope and fluorescence images captured with a Leica DFC450 C camera and processed by the Leica Application Suite Advanced Fluorescence software (LAS AF 3.1.0 build 8587).

### 2.6. Detection of IgG Antibodies to K. pneumoniae in Human Serum

Human serum specimens were tested for IgG antibodies to *K. pneumoniae* by micro-immunofluorescence as previously described [38]. Briefly, the *Klebsiella pneumoniae* HA391 strain transformed with the pDsRed plasmid (*K. pneumoniae* HA391 DsRed) expressing high levels of recombinant red fluorescent protein (RFP) (Addgene, Watertown, MA, USA) was grown in RPMI to OD_600_ = 0.35, followed by washing in PBS. Blood sera harvested as described above were diluted 1:100 in PBS, and bacteria were added to a final OD of 0.03, incubated at 37 °C for 30 min, and then washed three times in PBS, followed by resuspension in FITC-conjugated goat anti-human IgG, Fcγ-fragment-specific (Jackson ImmunoResearch, Cambridge, UK), diluted 1:200 in 0.1% BSA in PBS. The samples were incubated at 37 °C for 30 min and bacteria washed with PBS. Fluorescence microscopy analysis was carried out with a Leica DM4000 microscope.

### 2.7. Methylene Blue Staining of Bacterial Phagocytosis in Human Blood

*K. pneumoniae* HA391 DsRed was grown in RPMI to OD_600_ = 0.3 and 5 × 10^6^ cfu were resuspended in 500 µL of whole blood. After 60 min of incubation at 37 °C under shaking, one drop was smeared on a microscope glass slide, air-dried, and fixed with 4% paraformaldehyde for 20 min. Staining was carried out with 1% methylene blue for 1 min. Slides were washed in H_2_O and then air-dried. Phagocytosed bacteria were examined by contrast microscopy using a Leica DM4000 microscope.

### 2.8. Flow Cytometry Analysis of Bacterial Phagocytosis

Fresh anticoagulated whole blood samples were treated for 15 min with 10 µM DFP00173, 10 µM HTS13286, or the vehicle (1% DMSO) *K. pneumoniae* HA391 DsRed (5× *K. pneumoniae* HA391 strain 10^6^ cfu/250 µL of whole blood) and incubated at 37 °C for 0, 10, 30, or 60 min under shaking. After RBC lysis as detailed above, WBCs were pelleted by centrifugation (100× *g*, 5 min, at 4 °C) and washed with cold FACS buffer (PBS containing 0.5% BSA and 2 mM EDTA) and resuspended in PBS with the anti-CD14 APC labelled (BD Pharmingen, Franklin Lakes, NJ, USA) and anti-CD66b-v450-labelled (BD Pharmingen) antibodies at a dilution of 1:300 and 1:130, respectively. Fluorescence was analyzed with a CytoFLEX S flow cytometer (Beckman Coulter, Brea, CA, USA) equipped with three laser sources (405 nm, 561 nm, and 638 nm). Emitted fluorescent lights (FLs) were collected from specific fluorescent channels, with the blue fluorescence of (CD66b+) neutrophils in the FL1 channel (450/45 nm), the orange fluorescence of (RFP+) cells in the FL2 channel (585/42), and the red fluorescence of (CD14+) monocytes in the FL3 channel (712/25). The flow cytometry results were analyzed using FlowJo™ v10.8 Software (Becton, Dickinson, and Company).

### 2.9. Klebsiella Pneumoniae HA391 Survival Assay

Samples of fresh anticoagulated whole blood (450 µL) or PMNs (1.7 × 10^6^ cells in RPMI 1640 added to autologous serum at 1:1 volume ratio) were exposed or not (vehicle alone, 1% DMSO) to the AQP inhibitor (10 µM DFP00173, 10 µM HTS13286, or 10 µM RG100204) for 15 min at 37 °C in humidified atmosphere with 5% CO_2_. Cells were then infected by the addition of 50 µL *K. pneumoniae* HA391 DsRed (5 × 10^6^ cfu) grown overnight in Brain Heart Infusion (BHI) broth (Merck Group). After 10, 30, and 120 min of incubation at 37 °C under shaking, 30 µL aliquots collected from each infected cell suspension were lysed in 270 µL of cold PBS-0.1% saponin for 15 min. Thirty microliters of each lysed sample were diluted at a ratio of 1:10,000 in PBS and 20 µL of each corresponding dilution were plated in triplicate on Müller Hinton (MH) agar plates and incubated at 37 °C o/n. Colonies were counted manually and bacterial survival was expressed as percentage of the inoculum.

### 2.10. Transwell Assay of Cell Migration

Three hundred µL RPMI1640-Pen-Strept containing 5 × 10^5^ isolated PBMCs or PMNs were mixed with 300 µL autologous serum (50%) and different doses of DFP00173 or RG100204 or the vehicle alone (0.5% DMSO), before incubating for 15 min at 37 °C. Cell suspensions thus treated were transferred into 3 µm pore cell culture inserts (ThinCert, Greiner Bio-One, Kremsmunster, Austria) that were placed in a multiwell plate which had been prepared by adding 750 µL of RPMI1640-Pen-Strept, 750 µL of autologous serum, and the inhibitor or the vehicle alone in the lower compartment. Depending on the experimental condition, LPS at a final concentration of 100 ng/mL was added or not to the lower compartment solution. The multiwell plates thus prepared were incubated for 16 h at 37 °C. After the incubation, the cell culture insert and medium were removed from each well and cells that had migrated through the filter and attached themselves to the bottom of the well were fixed with 3.7% formalin for 10 min, permeabilized with 0.2% Triton X-100 for 7 min, and, finally, stained with DAPI (1 µg/mL) to be then counted by fluorescence microscopy.

### 2.11. Statistical Analysis

The expression levels of AQP3 and AQP9 in PMNs, neutrophil granulocytes, and PBMCs were compared by Student’s *t*-test. The role of AQP3 and AQP9 in bacterial phagocytosis, bactericidal activity, and cell migration was assessed using GraphPad Prism software (version 8.0.1). Data analysis was performed using two-way analysis of variance (ANOVA), followed by Tukey’s post hoc multiple-comparisons test. Statistical significance was defined as *p*-values less than 0.05.

## 3. Results

### 3.1. Expression and Localization of AQP3 and AQP9 in Human Leukocytes

The transcriptional expression and subcellular localization of AQP3 and AQP9 in samples of PMNs, neutrophil granulocytes alone, and PBMCs isolated from the peripheral blood of healthy male subjects were determined by RT-qPCR and immunofluorescence, respectively.

After normalization relative to the housekeeping gene *β-actin*, high levels of *AQP9* mRNA were found in all three WBC fractions that were examined (Figure 1A–C). The *AQP9* mRNA was particularly high in PMNs, especially in neutrophil granulocytes. *AQP3* was expressed in PBMCs (Figure 1C), where it was more abundant than *AQP9,* while it was barely detectable in all neutrophil types (Figure 1A,B). Expression analysis was also extended to AQP1 and AQP5 in consideration of previous works reporting on AQP5 in neutrophil granulocytes [22,39] and moderate levels of AQP1 and AQP5 in THP-1 cells, a human monocyte cell line [40]. Both AQP1 and AQP5 were barely detectable or absent in all WBC fractions when compared to the levels of *AQP3* and *AQP9* (Figure 1A–C).

For the immunofluorescence analysis, aliquots of isolated neutrophil granulocytes and PBMCs were incubated with anti-AQP9 or anti-AQP3 polyclonal antibodies and processed as reported in Materials and Methods. Cell nuclei were counterstained with DAPI (Figure 2 and Figure 3; blue fluorescence). Consistent with the results of the RT-qPCR, a strong AQP9 immunofluorescence was seen in neutrophil granulocytes. AQP9 immunoreactivity was observed both at the cell membrane and in the intracellular compartment where it appeared to be granular (Figure 2E,F; green fluorescence), whereas no immunoreactivity was seen with the AQP3 antibody (Figure 2B,C). The specificity of the anti-AQP9 and anti-AQP3 antibodies was tested on Jurkat cells, an immortalized human T lymphocyte cell line expressing AQP9 [41], and on human keratinocytes, a cell type expressing AQP3 [42,43], respectively. Negative controls were carried out by omitting the primary antibody (Appendix A).

Among PBMCs, both monocytes and lymphocytes showed significant AQP3 immunoreactivity, with the former apparently slightly more fluorescent than the latter (Figure 3B,C and Figure 3E,F, respectively). With the AQP9 antibodies, both monocytes and lymphocytes showed strong and comparable immunostaining over the plasma membrane and at the intracellular compartment (Figure 3H,I).

### 3.2. Inhibition of AQP3 or AQP9 Do Not Impair the White Blood Cell Phagocytosis and Killing of the Gram-Negative Pathogen Klebsiella pneumoniae

Given the suggested involvement of AQP3 and AQP9 in infection, inflammation, and the development of disease [14], we investigated the contribution of these two AQPs to leukocyte phagocytosis.

The potential involvement of AQP3 and AQP9 in the leukocyte phagocytosis of infecting bacteria was investigated with a strain of the Gram-negative pathogen *K. pneumoniae* expressing a recombinant red fluorescent protein (*K. pneumoniae* HA391 DsRed) allowing the visualization of the whole bacteria. The *K. pneumoniae* HA391 DsRed was also used to run the subsequent bacterial killing studies.

Firstly, we worked to rule out the risk that the phagocytic immune response to the *K. pneumoniae* infection could depend completely on the innate immunity of the blood sample donors rather than on the contribution of the pre-existing presence of antibodies directed against *K. pneumoniae* (adaptive immunity). Aliquots of each serum specimen were, therefore, incubated with the *K. pneumoniae* strain, and an FITC-conjugated antibody directed against the human immunoglobulins G was used to detect the eventual presence of anti-*K. pneumoniae* antibodies (see *Materials and Methods* for details). For each blood sample, a negative control condition was run by omitting the blood serum. No antibodies against *K. pneumoniae* were detected in all serum samples analyzed, suggesting that the immune recognition of the *K. pneumoniae* strain was not serum-mediated (Figure 4A–D).

The role of AQP3 and AQP9 in the phagocytosis of *K. pneumoniae* HA391 DsRed was monitored over a 60 min time course. Blood samples exposed to the BHI medium served as controls. Thereby, bacterial ingestion was accompanied by distinct changes in the cell size and morphology compared to neutrophils exposed to the medium alone (Figure 5A,B). This can be explained by the bacterial LPS that induces a better adherence of granulocytes and, thereby, leads to changes in the infected granulocytes [44].

In a series of flow cytometry experiments, neutrophil and monocyte phagocytosis with *K. pneumoniae* HA391 DsRed was carried out by the selective inhibition of AQP3 or/and AQP9. Whole blood samples were incubated with BHI medium infected or not with *K. pneumoniae* for 10, 30, or 60 min. For each time point, aliquots of the infected blood samples received the inhibitor vehicle alone (1% DMSO), the AQP3 or/and the AQP9 inhibitor (10 µM DFP00173 and 10 µM HTS13286, respectively), or neither the vehicle nor the AQP blockers (basal condition). After erythrocyte lysis and washout, blood sample specimens were submitted to a flow cytometry evaluation of the RFP-positive neutrophils or monocytes, depending on the specific cell marker that was employed, CD66b for neutrophils and CD14 for monocytes. The percentages of RFP-positive neutrophils (Figure 6B–D) or monocytes (Figure 7B–D) were calculated based on the fluorescence intensity of the internalized RFP after arbitrarily setting the fluorescence of the uninfected blood samples to 0% (Figure 6A and Figure 7A). After 60 min of inoculation with *K. pneumoniae,* about 32% and 43%, respectively, of neutrophils and monocytes, respectively, had ingested the pathogen (Figure 6D and Figure 7D). At all three time points, the addition of neither DFP00173 nor HTS13286, whether individually or in combination, impaired the phagocytosis of *K. pneumoniae* HA391 DsRed by neutrophils (Figure 6E–G and Figure 7E–G) or monocytes, with one exception: both blockers used together for 60 min led to a reduction in infected monocytes by approximately 40% (Figure 7G).

Experiments were then conducted to investigate if AQP3 and/or AQP9 could contribute to the resolution of the phagocytic process through which the ingested bacterium is degraded by the lysosomes. A first series of experiments was performed with aliquots of the whole blood samples. A survival curve was constructed by exposing the blood samples to the BHI medium infected with *K. pneumoniae* for up to 120 min. Live intracellular bacteria were quantified after plating serial dilutions of the lysate of the various whole blood samples on LB agar plates. After 120 min, all infecting bacteria were killed by blood phagocytes with a killing score of 65% already achieved after the first 10 min of infection (Figure 8A). The involvement of AQP3 and AQP9 in bacterial killing was evaluated by measuring the percentage of bacterial survival at 10, 30, and 120 min of infection in the presence or not of 10 µM DFP00173 or HTS13286 or both inhibitors added together (Figure 8B). Infected blood samples without the addition of the vehicle or inhibitors, or with the vehicle alone were used as controls (Figure 8B; black and grey histograms). The inhibition of AQP3 or AQP9 or both at the same time did not impair the killing efficacy of leukocytes.

A second series of experiments investigated the specific involvement of AQP9 in the bacterial killing by PMNs. In addition to HTS13286, RG100204, a novel specific inhibitor of AQP9 [18,34], was used. Isolated PMNs were infected with the *K. pneumoniae* bacteria for up to 120 min, and the surviving intracellular bacteria were quantified at the time points of 10, 30, and 120 min (Figure 9A). After 120 min of infection, nearly 90% of bacteria were eliminated by PMNs. No bacterial killing impairment was seen at all three time points in the PMNs that were exposed to 10 µM RG100204 compared to the related controls where the vehicle was omitted or compared to the vehicle alone (Figure 9B).

The AQP3 inhibitor DFP00173 was not tested, as the RT-qPCR analysis did not detect a substantial expression of AQP3 in PMNs (Figure 1A).

### 3.3. Selective Inhibition of AQP3 and AQP9 Reduces Cell Migration of PBMCs and Neutrophils

Next, we investigated the role of AQP3 and AQP9 in the migration of PMNs and PBMCs in response to the pathogen-associated molecular pattern (PAMP) LPS. A transwell assay was used to evaluate the migration of the WBC specimens in the presence or absence of DFP00173 or RG100204.

With PMNs, RG100204 was used at a single dose (10 µM) in the transwell migration experiments carried out in the absence of LPS, while, in the evaluations made in the presence of LPS, the inhibitor was applied in a range of scalar doses, from 10 µM up to 14 nM. Compared to the conditions with the vehicle alone (0.5% DMSO), RG100204 significantly reduced the migration of PMNs through the transwell filter both in the absence and presence of LPS. The lowest concentration of RG100204 that conferred a significant reduction in LPS-stimulated cell migration was 41 nM (Figure 10 and Figure 11).

Since PBMCs express both AQP3 and AQP9 (Figure 1), cell migration experiments with these cells were performed using both inhibitors, separately, for 16 h both under an LPS challenge and in its absence. Interestingly, RG100204 was found to impair the cell migration of human PBMCs both with and without an LPS challenge, while DFP00173 led to a significant reduction in PBMC migration only under LPS stimulation (Figure 12).

## 4. Discussion

AQPs are emerging as important players at various levels in immunity, a process in which WBCs and other specialized cells of the immune system mediate protection against pathogens to maintain homeostasis. In this work, we investigated the expression and involvement of AQP3 and AQP9, two aquaglyceroporins also permeable to hydrogen peroxide, in important processes of immunity such as cell migration, and the phagocytosis and killing of ingested bacteria by WBCs following infection.

The RT-qPCR and immunofluorescence analyses showed the expression of AQP3 in monocytes and, to a lower extent, in lymphocytes. In PMNs instead, the level of AQP3 was barely detectable. There, AQP9 is highly expressed, especially in neutrophil granulocytes. AQP9 was also found in monocytes and lymphocytes. This profile is consistent with previous studies by other authors [11,17,45,46]. In humans, neutrophils have also been reported to express AQP5 [22], while AQP1 and AQP5 were detected in B and T lymphocytes together with AQP3 and have been reported to be upregulated after cell activation [45]. Therefore, the low expression of AQP3 seen in lymphocytes may be explained by the basal conditions in which the PBMCs were evaluated in the current study. The AQP9 immunoreactivity seen at the intracellular level suggests that the translocation of this AQP to the plasma membrane is regulated, in line with a previous work reporting that the phosphorylation of AQP9 at S11 through a Rac1-dependent pathway mediates membrane localization and neutrophil polarization [47].

The specificity of DFP00173 to AQP3, HTS13286 to AQP9, and RG100204 to AQP9, compared to the most homologous AQP isoforms, has been established previously [18,32,33]. Furthermore, selectivity has not been formally established. However, phenotypes induced by the inhibitors in various test systems are similar to the effects induced by an AQP3-targeting antibody [48], and by *Aqp9* gene deletion [16,18,24].

The availability of these inhibitors allowed us to evaluate the individual roles of AQP3 and AQP9 in various functions of innate immunity. The contributions of AQP3 and AQP9 to the phagocytosis and killing of pathogenic bacteria were studied in vitro using an engineered *K. pneumoniae* strain. No apparent involvement of individually targeted AQP3 or AQP9 was observed in the phagocytosis or phagolysosomal degradation of *K. pneumoniae* by neutrophils or monocytes. Interestingly, the significant impairment of the phagocytic capacity of monocytes was observed only when both AQPs were inhibited simultaneously. No functional involvement of either AQP3 or AQP9 in bacterial clearance was observed. Regarding AQP9 in neutrophils, this result was a bit surprising, since AQP9 was suggested to have a key role in the polarization of primary human neutrophils [46].

Neither AQP9 nor AQP3 appear to play an individual role in the bacterial phagocytosis by monocytes. However, a significant decrease in the phagocytosis of *K. pneumoniae* was observed when both AQPs were inhibited simultaneously. This interesting result is in line with the previous work where human macrophages were infected with *P. aeruginosa* [5]. The important involvements of AQP3 and AQP9 have already been reported, in terms of the changes in cell volume and morphology and in the phagocytic function of human and rodent macrophages [4,5,49]. However, it should also be noted that AQP redundancy in human monocytes may include two additional AQPs, AQP1 and AQP5, that were detected at moderate levels in THP-1 cells [40]. In this study, in all three WBC fractions, both AQP1 and AQP5 were barely detectable. Hence, further research is needed in order to understand signaling pathways that are associated with each of these channels in leukocytes.

On the other hand, both AQP3 as well as AQP9 were found to be involved in leukocyte migration. Thereby, AQP9 seems to contribute to PMN migration at the basal level, but also when migration was stimulated with LPS, whereas AQP3 affected PBMC migration only when cells were challenged with LPS. Therefore, the active function AQP9 played in regulating human neutrophil volume may be necessary for activities of cell motility such as those performed in extravasation and/or chemotaxis to move towards potentially harmful microorganisms.

The key roles for AQP3 and AQP9 in immune cell migration have been repeatedly reported both in innate and adaptive immunity (see [11,14] for reviews). However, to our knowledge, this is the first study in which the individual contribution of each one of these channels in immune cell migration is evaluated using specific inhibitors. We show that the selective inhibition of AQP9 with RG100204 strongly impairs neutrophil cell migration in vitro. This is fully consistent with previous studies using murine models of hyperinflammation where AQP9 deletion or inhibition by RG100204, respectively, led to a significant reduction in neutrophil tissue infiltration [7,18,26]. Here, we observed that neutrophil cell migration was reduced by approximately 62% and 68%, respectively, after the specific inhibition of AQP9 with or without LPS stimulation (Figure 11). This suggests an important role for AQP9 in neutrophil migration. A role for an additional AQP channel, AQP5, in neutrophil locomotion was suggested after observing the impaired neutrophil tissue infiltration and increased survival in severe sepsis in subjects with srs3759129 polymorphism, which was accompanied by reduced neutrophil AQP5 [21,22,50]. Here, we find that the expression level of AQP5 in isolated human neutrophils is barely appreciable. Further work is, therefore, needed to unravel this apparent discrepancy.

Moreover, we observed that the selective inhibition of AQP9 impairs LPS-stimulated and unstimulated PBMC migration, while the inhibition of AQP3 only reduced LPS-induced migration (Figure 12). Taken together, these data suggest that AQP3 and AQP9 contribute to the migration of PBMCs in distinct ways. This seems to be consistent with a recent study indicating opposing roles for leukocyte AQPs 3 and 9 in human sepsis [20]. High levels of AQP3 were associated with augmented survival, whereas AQP9 did not appear to change during sepsis when normalized against the neutrophil count, and a low expression of AQP9 was associated with increased survival. The neutrophil AQP9 expression was considered an independent risk factor for sepsis lethality.

However, further studies are needed to better understand which molecular transport facilitated by AQP3 and AQP9 is affecting downstream signaling and leading to altered locomotion. The movement of water could be related to the cell volume/morphology, cytoskeletal rearrangement, dynamical adhesion, and retraction and motility (i.e., filopodia formation), and the movement of glycerol could be related to metabolic shifts that could activate some immune functions, while the transport of hydrogen peroxide could reasonably be linked to the activation/modulation of some signaling pathways such as NF-κB and p38 MAPK, leading to the activation of inflammasomes (i.e., NLRP3) and the consequent production of proinflammatory interleukins [7,16,18,19,24,51,52]. The AQP3-mediated H_2_O_2_ uptake was found to be required for chemokine-dependent T-cell migration through the activation of the Rho family GTPase Cdc42 and the subsequent actin dynamics [53]. The availability of the specific and potent inhibitors of AQP3 and AQP9 now allows us to answer these questions step by step. Unraveling the precise molecular mechanisms by which AQPs affect cell motility remains challenging. In part, this may be due to the generally low expression of AQPs in dedifferentiated cell lines [54].

## 5. Conclusions

In conclusion, in human WBCs, AQP9 is expressed both in neutrophils where it is particularly abundant, and in PBMCs, whereas AQP3 is found only in the latter, with a higher expression in monocytes than in lymphocytes. AQP9 plays major roles in neutrophil and PBMC basal locomotion, and chemotaxis. AQP3 impacts monocyte chemotaxis. The phagocytosis of *K. pneumoniae* is only impaired when both AQPs are blocked simultaneously. The two AQPs do not appear to have roles in bacterial killing. Despite a role in neutrophil locomotion that might reduce innate immunity, we have previously shown that the inhibition of AQP9 can be beneficial in a murine model of bacterial sepsis [18].

The results of the present work provide further and useful insights into the role of AQPs in innate immunity and their relevance as markers and potential therapeutic targets of inflammatory diseases.

## Figures and Tables

**Figure 1 cells-14-00880-f001:**
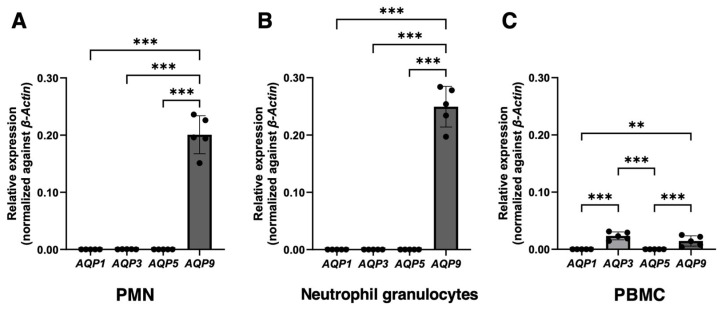
Real-time quantitative PCR analysis of *AQP1*, *3*, *5,* and *9* mRNA expression in human white blood cells. PMNs, neutrophil granulocytes alone, and PBMCs were isolated from the peripheral blood of healthy donors (*n* = 5). The mRNA levels of the four AQPs were normalized against those of the housekeeper gene *β-Actin*. High levels of *AQP9* mRNA are detected in polymorphonuclear leukocytes (**A**), especially in neutrophil granulocytes (**B**). The level of *AQP9* transcript is considerably less abundant in PBMCs (**C**). *AQP3* is expressed in PBMCs (**C**), while, in neutrophils, it is very weak or absent (**A**,**B**). *AQP1* and *AQP5* are barely appreciable or absent in all three WBC fractions (**A**–**C**). Data are shown as mean ± SEM (*n* = 5). **, *p* < 0.01; ***, *p* < 0.001.

**Figure 2 cells-14-00880-f002:**
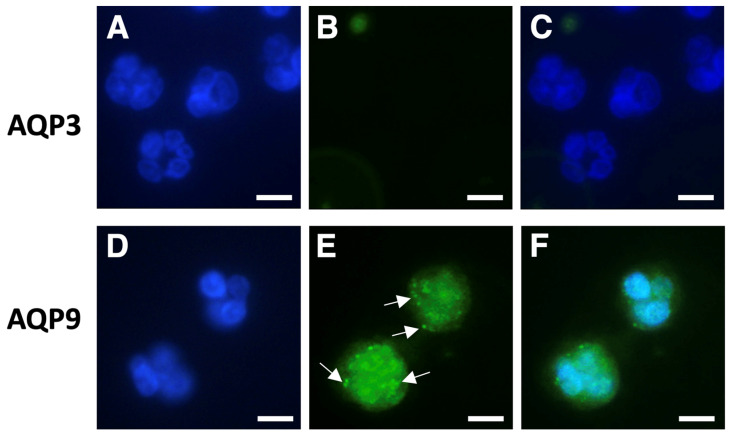
Immunofluorescence analysis of AQP3 and AQP9 in isolated human neutrophils. Neutrophils were isolated by immune-magnetic negative selection from the peripheral blood of healthy donors and analyzed using immunofluorescence with anti-AQP3 or anti-AQP9 polyclonal antibodies ((**B**) and (**E**), respectively). Cell nuclei were counterstained with DAPI ((**A**,**D**); blue fluorescence). (**C**) and (**F**) represent the merging of (**A**,**B**) and (**D**,**E**), respectively. Strong AQP9 immunofluorescence is seen in neutrophils, where it appears to be granular (arrows), and over the plasma membrane (**E**); green fluorescence). By contrast, neutrophils do not show any AQP3 immunoreactivity (**B**,**C**). Scale bars, 10 µm.

**Figure 3 cells-14-00880-f003:**
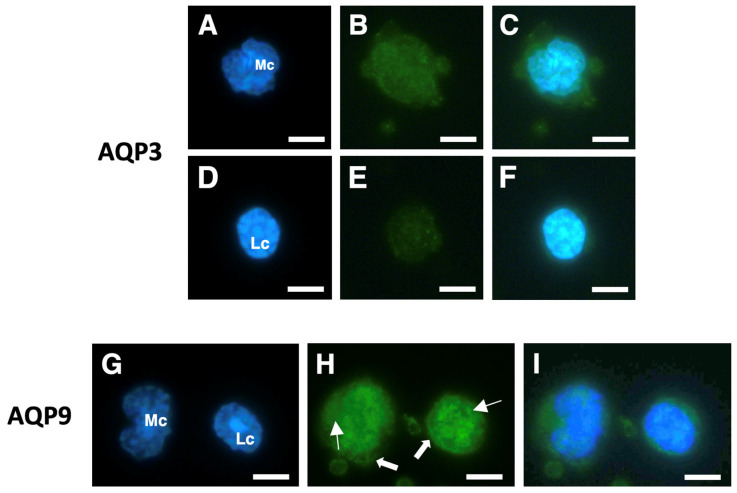
Immunofluorescence analysis of AQP3 and AQP9 in isolated human PBMCs. Isolated PBMCs were subjected to immunofluorescence (green) with anti-AQP3 (**B**,**E**) or anti-AQP9 (**H**) polyclonal antibodies. Cell nuclei were stained with DAPI ((**A**,**D**,**G**); blue fluorescence). Cells in (**C**), (**F**), and (**I**) are the merging of (**A**,**B**), (**D**,**E**), and (**G**,**H**), respectively. AQP3 immunoreactivity is high in monocytes (**B**,**C**), while it is barely appreciable in lymphocytes (**E**,**F**). Strong AQP9 immunofluorescence is instead seen in monocytes and lymphocytes both at the intracellular compartment (solid arrows) and over the plasma membrane (linear arrows) (**H**,**I**). Mc, monocyte; Lc, lymphocyte. Scale bars, 10 µm.

**Figure 4 cells-14-00880-f004:**
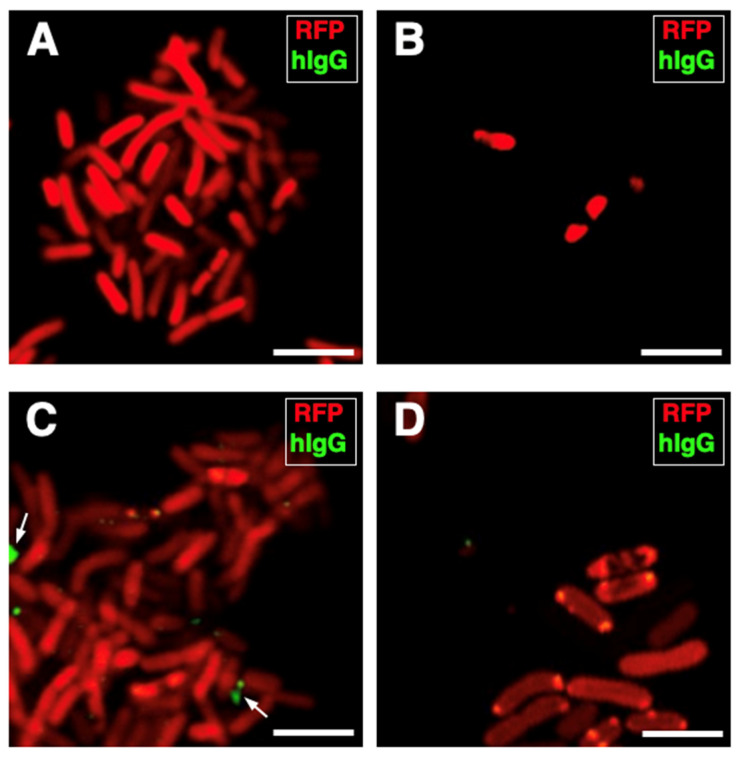
Human serum specimens are without antibodies to *K. pneumoniae* HA391. *K. pneumoniae* HA391 DsRed was incubated with the blood sera of three of the five subjects used for the study. An FITC-conjugated antibody against the human immunoglobulins G (hIgG; green fluorescence) was used to detect eventual anti-*K. pneumoniae* antibodies. (**A**) Negative control (absence of serum). (**B**–**D**) *K. pneumoniae* HA391 DsRed incubated with each one of the three sera, separately. Arrows indicate unspecific spots located outside of bacterial bodies (**C**). Scale bars, 2 µm.

**Figure 5 cells-14-00880-f005:**
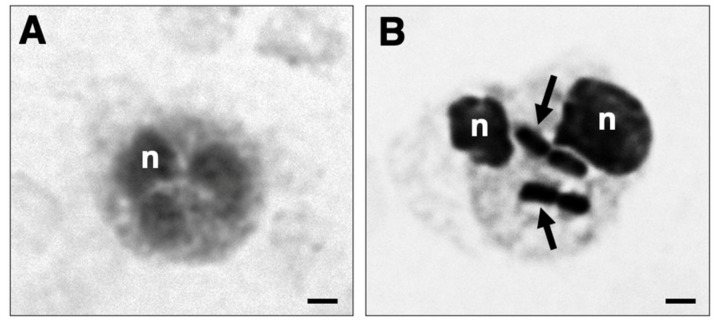
Phagocytosis of *K. pneumoniae* HA391 DsRed by blood neutrophils. Black and white micrograph of neutrophils in whole blood, stained with methylene blue. (**A**) Neutrophil from a blood sample incubated for 60 min with BHI medium. (**B**) Phagocytic-positive neutrophil from a blood sample infected for 60 min with *K. pneumoniae* HA391 DsRed in BHI medium. Arrows indicate the ingested bacteria. A distinct change in cell size and morphology is noted compared to the uninfected neutrophil shown in (**A**). n, nucleus. Scale bars, 2 µm.

**Figure 6 cells-14-00880-f006:**
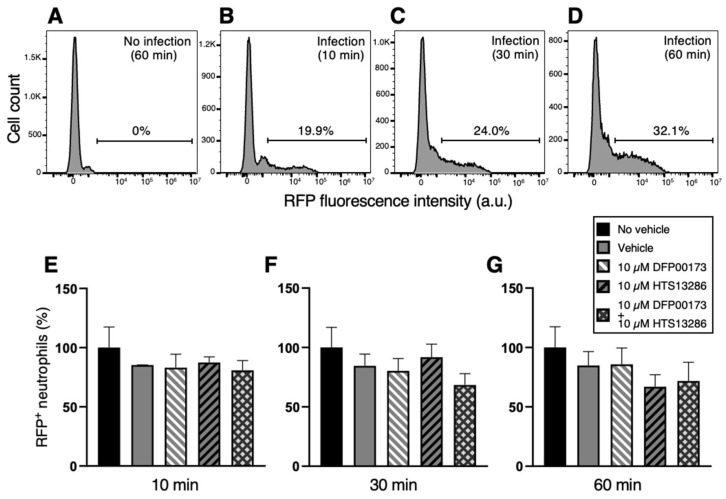
Flow cytometry analysis of neutrophil phagocytosis of *K. pneumoniae* after AQP3 and/or AQP9 inhibition. Whole blood was exposed to *K. pneumoniae* HA391 DsRed, or BHI control medium (with or without vehicle) for 10, 30, or 60 min. After red blood cells lysis and washout, flow cytometry analysis of neutrophils was conducted by selection through the CD66b marker. Percentages of RFP-positive neutrophils were calculated. (**A**) Basal condition. Blood sample exposed to the medium alone for 60 min (no infection). (**B**–**D**) Blood samples infected with *K. pneumoniae* for 10, 30, or 60 min, respectively. About one third of neutrophils are infected after 60 min of exposure to *K. pneumoniae* HA391 DsRed (**D**). (**E**–**G**) Percentage of RFP-positive in the different control and experimental conditions (inset), following 10, 30, or 60 min of infection, respectively. There are no statistically significant differences among the different conditions. Vehicle, 1% DMSO. Percentages are expressed as mean ± SEM (*n* = 5). a.u., arbitrary units.

**Figure 7 cells-14-00880-f007:**
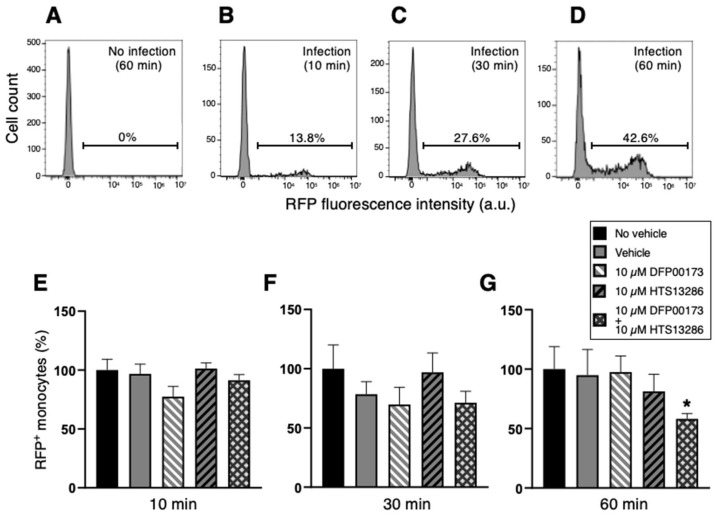
Flow cytometry analysis of monocyte phagocytosis of *K. pneumoniae* after AQP3 and/or AQP9 inhibition. Whole blood was infected (or not) with BHI medium containing *K. pneumoniae*. After washout and red blood cell lysis, flow cytometry analysis of monocytes was conducted by selection through the CD14 marker. Percentages of phagocytic-positive monocytes were calculated through the signal of the internalized red fluorescent protein (RFP). (**A**) Basal condition. Blood sample exposed to the medium alone for 60 min (no infection). (**B**–**D**) Blood samples infected with *K. pneumoniae* for 10, 30, or 60 min, respectively. More than 40% of monocytes are infected after 60 min exposure to the bacterial strain (**D**). (**E**–**G**) Percentage of phagocytic-positive monocytes (RFP^+^) in the different control and experimental conditions (see inset) following 10, 30, or 60 min of infection, respectively. The only statistically significant difference is seen between the control with no vehicle and the experimental condition where the two inhibitors, DFP00173 and HTS13286, were both added. Vehicle, 1% DMSO. Percentages are expressed as mean ± SEM (*n* = 5). *, *p* < 0.05.

**Figure 8 cells-14-00880-f008:**
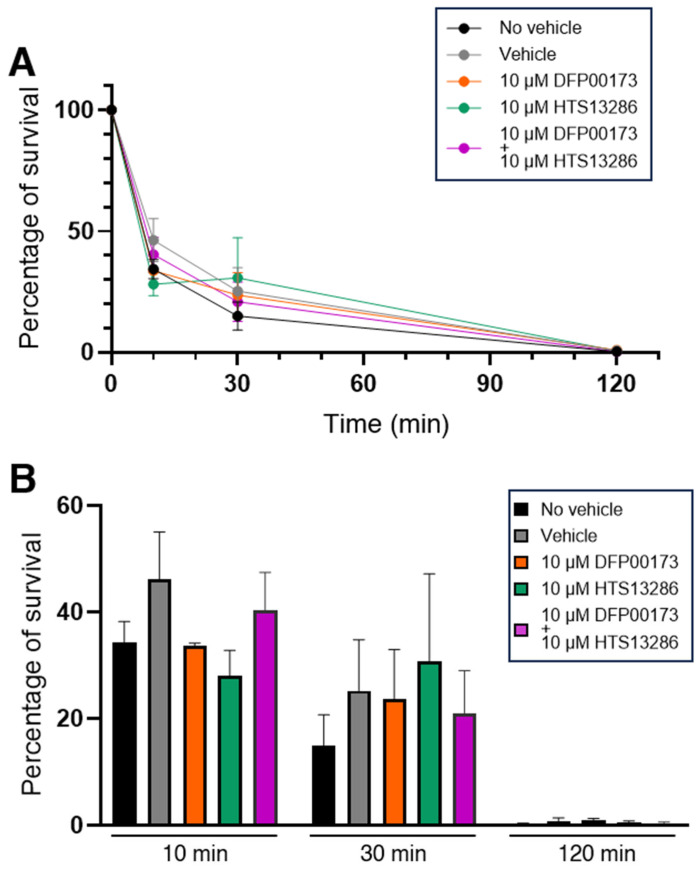
Inhibition of AQP3 and AQP9 does not impair killing of *K. pneumoniae* by WBCs. Whole blood samples were exposed to the BHI medium containing *K. pneumoniae* for up to 120 min and intracellular bacteria were quantified by lysis, serial dilution, and viable counting on LB agar plates (see Materials and Methods for details). (**A**) Survival curves of *K. pneumoniae* inoculated in whole blood samples and evaluated after 10, 30, and 120 min of incubation in absence or presence of vehicle (1% DMSO) and AQP3/AQP9 selective inhibitor. In all conditions, all inoculated bacteria are almost totally eliminated within 120 min. (**B**) Percentages of bacterial survival in presence or absence of the AQP3 or AQP9 inhibitor after 10, 30, or 120 min of exposure to whole blood. Both DFP00173 and HTS13286 do not impair the clearance efficacy of leukocytes. Percentages are expressed as mean ± SEM (*n* = 5).

**Figure 9 cells-14-00880-f009:**
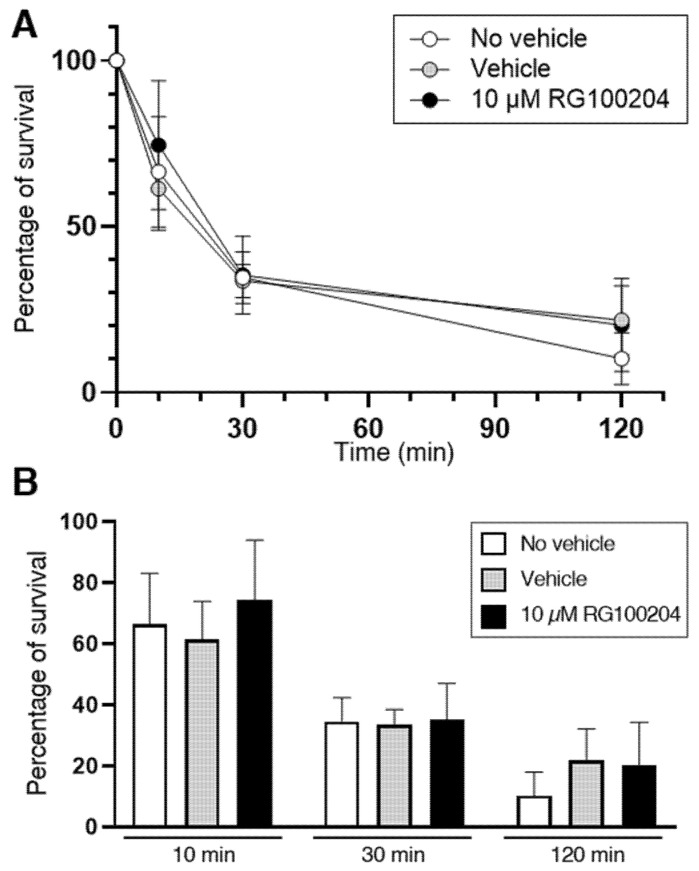
Killing of *K. pneumoniae* by PMNs is not impaired by the AQP9 inhibitor RG100204. Isolated PMNs were infected with *K. pneumoniae* for up to 120 min and surviving intracellular bacteria were quantified by lysis, serial dilution, and viable counting on LB agar plates (see Materials and Methods for details). (**A**) Survival curve of *K. pneumoniae* inoculated in PMN samples and analyzed after 10, 30, and 120 min of incubation in absence or presence of vehicle or 10 µM RG100204. In all three conditions, after 120 min of incubation, nearly 90% of bacteria are resolved by PMNs. (**B**) Percentages of bacterial survival in presence or absence (see inset for conditions) of the AQP9-selective inhibitor RG100204 after 10, 30, or 120 min of killing by PMNs. RG100204 does not seem to affect the killing capacity of PMNs. Percentages are expressed as mean ± SEM (*n* = 5).

**Figure 10 cells-14-00880-f010:**
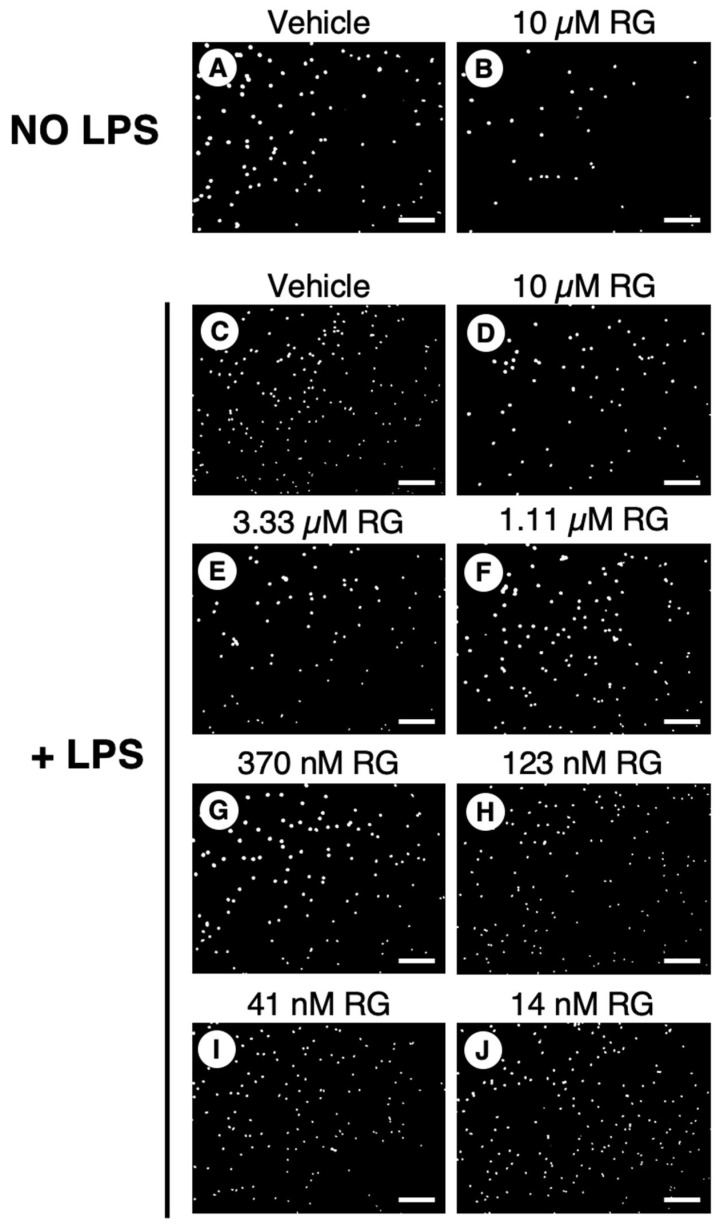
Effect of RG100204 on LPS-induced cell migration of human PMNs. Representative micrographs showing the PMNs that had migrated (white spots) across the transwell filter after 16 h with (panels (**C**–**J**)) or without (panels (**A**,**B**)) LPS challenging in absence (vehicle, 0.5% DMSO; panel (**C**)) or presence of a series of doses of the AQP9 inhibitor RG100204 (RG; panels (**D**–**J**)). Scale bars, 150 µm.

**Figure 11 cells-14-00880-f011:**
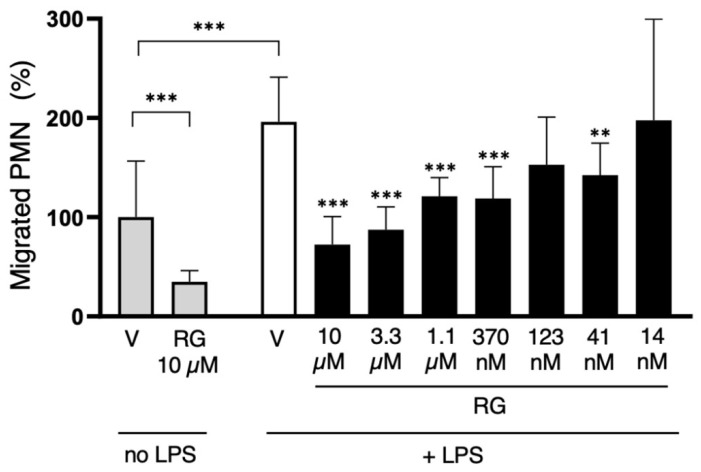
The AQP9-selective inhibitor RG100204 reduces neutrophil migration. Migration analysis of isolated whole PMNs after 16 h with or without LPS challenge, and in presence or absence (vehicle alone, 0.5% DMSO; V) of serial concentrations of RG100204 (RG). Transmigrated PMNs are expressed as percentage, compared to the number of PMNs migrating without LPS stimulation and in the presence of the vehicle alone. Percentages are expressed as mean ± SEM (*n* = 5; 6 fields/condition). ** *p* < 0.01; *** *p* < 0.001.

**Figure 12 cells-14-00880-f012:**
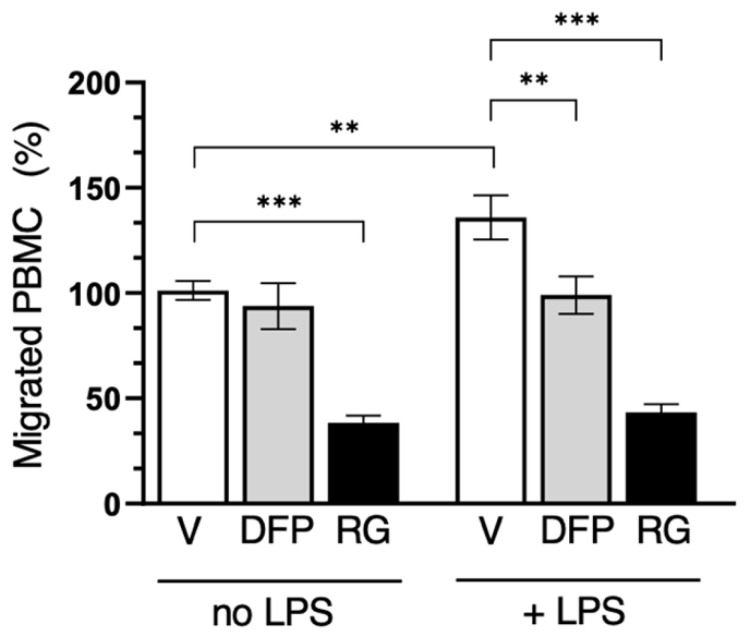
Selective inhibition of both AQP3 and AQP9 reduces PBMC migration. Migration analysis of isolated PBMCs after 16 h with or without LPS challenge and in absence (vehicle alone, 0.5% DMSO; V) or presence of 10 µM DFP00173 (DFP) or 10 µM RG100204 (RG). Migrated PBMCs are expressed as percentage of PBMCs migrating without LPS stimulation and in presence of vehicle alone. Percentages are expressed as mean ± SEM (*n* = 5; 10 fields/condition). ** *p* < 0.01; *** *p* < 0.001.

## Data Availability

The original contributions presented in this study are included in the article. Further inquiries can be directed to the corresponding authors.

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
