# Peer review of "Selective Blockade of Two Aquaporin Channels, AQP3 and AQP9, Impairs Human Leukocyte Migration"

_cells, 2025, doi:10.3390/cells14120880_

Round 1
Reviewer 1 Report
Comments and Suggestions for Authors
The research article by Sabino Garra et al. investigates the role of aquaporins AQP3 and AQP9 in human leukocyte function, particularly in cell migration, phagocytosis, and bacterial death during an immune response against infection caused by Klebsiella pneumoniae. The main findings of the study suggest distinct and critical contributions of AQP3 and AQP9 to leukocyte motility, but no significant involvement in bacterial killing. AQP3 and AQP9 play distinct and critical roles in leukocyte motility, but are not directly involved in bacterial killing. Targeting these AQPs could offer therapeutic potential in modulating immune responses without compromising microbial clearance. This study provides valuable insights into an underexplored aspect of human immune cell function and highlights the role of aquaporins in cell motility. The manuscript presents an innovative perspective on peripheral blood leukocytes. However, the precise molecular targets, signaling pathways, and mechanisms through which AQP3 and AQP9 regulate cellular motility remain undefined. Addressing these mechanistic gaps would substantially enhance the impact and translational relevance of the study, potentially offering novel therapeutic strategies. Below are the major and minor concerns that should be addressed by the authors:
1st - Major concerns
1 - Mechanistic Role of AQP3 and AQP9: The study lacks data demonstrating the distinct roles of AQP3 and AQP9 in human leukocyte migration. No direct evidence is provided linking AQP function to cytoskeletal dynamics, volume changes, or signaling alterations. To strengthen the mechanistic depth and translational impact, I suggest expanding functional assays to dissect the downstream signaling affected by AQP inhibition. Live-cell imaging to monitor cytoskeletal rearrangements upon AQP inhibition; cell volume assays to assess if AQP inhibition affects dynamic volume regulation during migration; and/or ROS measurement assays to explore the role of AQP-mediated Hâ‚‚Oâ‚‚ transport in signaling could be performed to strength the study and provide potential mechanisms.
2 - Other Aquaporins (AQP1, AQP5): The authors mention possible redundancy with AQP1 and AQP5 but do not test it experimentally. Pharmacological inhibitors might not fully rule out redundancy. Expression profiling assays (e.g., qPCR, Western blot, or RNA-seq) should be performed to check if inhibiting AQP3 and/or AQP 9 leads to upregulation of other AQPs (AQP1, AQP5).
3 - Timepoints of Bacterial Killing: The authors showed that AQP inhibition does not impair killing is important, however this could based on limited timepoints. The authors could provide data for longer-term bacterial survival (more than 1hour). I suggest to extend bacterial killing assays to longer timepoints (e.g., 2-4 hours).
2nd - Minor concerns
A - Figure 1: Provide the same calibration scale for the graphics on panel A, B and C. A table of the primer sequence used should be provide as supplementary material. Justify why the β-actin was used as reference instead other house-keep gene.
B - Figure 2 and 3: the authors should include the positive and negative controls for primary and secondary antibody autofluorescence. The images should be provided as supplementary material.
C - Figure 8 and 9: In the panel A of the figure 8 and figure 9, the author should provide survival curve of K. pneumoniae for all the experimental groups (vehicle, 1% DMSO, and AQP3/AQP9 selective inhibitors).
D - Figure 10: Provide a calibration scale for the representative micrographs of the on LPS-induced cell migration assay.
E - AQP3 vs. AQP9: It’s not clear how specific the inhibitors are for AQP3 vs. AQP9. Are off-target effects ruled out? A short discussion should be included on the relevance of AQP3/AQP9 selective inhibitors.
F - Selectivy: Include data or discussion validating the selectivity of DFP00173, HTS13286, and RG100204.
G - Mechanistic Depth: The article lacks mechanistic insight into how AQP3/AQP9 affect migration. Is it through cytoskeletal rearrangement, volume regulation, or signaling modulation? Discussion regarding the potential pathways could be provide.
H - Discussion: The authors could provide more insights and explore more therapeutic implications. Can AQP inhibition modulate pathological inflammation without impairing host defense?
Author Response
Reviewer 1
Comments and Suggestions for Authors
The research article by Sabino Garra et al. investigates the role of aquaporins AQP3 and AQP9 in human leukocyte function, particularly in cell migration, phagocytosis, and bacterial death during an immune response against infection caused by Klebsiella pneumoniae. The main findings of the study suggest distinct and critical contributions of AQP3 and AQP9 to leukocyte motility, but no significant involvement in bacterial killing. AQP3 and AQP9 play distinct and critical roles in leukocyte motility, but are not directly involved in bacterial killing. Targeting these AQPs could offer therapeutic potential in modulating immune responses without compromising microbial clearance. This study provides valuable insights into an underexplored aspect of human immune cell function and highlights the role of aquaporins in cell motility. The manuscript presents an innovative perspective on peripheral blood leukocytes. However, the precise molecular targets, signaling pathways, and mechanisms through which AQP3 and AQP9 regulate cellular motility remain undefined. Addressing these mechanistic gaps would substantially enhance the impact and translational relevance of the study, potentially offering novel therapeutic strategies. Below are the major and minor concerns that should be addressed by the authors:
1st - Major concerns
1 - Mechanistic Role of AQP3 and AQP9: The study lacks data demonstrating the distinct roles of AQP3 and AQP9 in human leukocyte migration. No direct evidence is provided linking AQP function to cytoskeletal dynamics, volume changes, or signaling alterations. To strengthen the mechanistic depth and translational impact, I suggest expanding functional assays to dissect the downstream signaling affected by AQP inhibition. Live-cell imaging to monitor cytoskeletal rearrangements upon AQP inhibition; cell volume assays to assess if AQP inhibition affects dynamic volume regulation during migration; and/or ROS measurement assays to explore the role of AQP-mediated Hâ‚‚Oâ‚‚ transport in signaling could be performed to strength the study and provide potential mechanisms.
Answer. We thank the Reviewer for this comment. Unravelling the precise molecular mechanisms through which, distinctly, AQP3 and AQP9 intervene in the migration of human leukocytes remains challenging. As specified at the end of the Discussion since the first submission of the manuscript, this task will be the object of future studies for which we are currently equipping ourselves with specifically immortalized promyelocytic and myeloid progenitor cell lines. As written in the discussion, the availability of specific and potent inhibitors of AQP3 and AQP9 allows now to answer these questions step by step.
2 - Other Aquaporins (AQP1, AQP5): The authors mention possible redundancy with AQP1 and AQP5 but do not test it experimentally. Pharmacological inhibitors might not fully rule out redundancy. Expression profiling assays (e.g., qPCR, Western blot, or RNA-seq) should be performed to check if inhibiting AQP3 and/or AQP9 leads to upregulation of other AQPs (AQP1, AQP5).
Answer. We thank the reviewer for the question. We evaluated the expression of AQP1 and AQP5 in all three WBC fractions (PMN, neutrophil granulocytes and PBMC) finding them barely appreciable compared to AQP3 and AQP9 if not even absent (Figure 1). As done for AQP3 that by RT-qPCR in basal conditions did not show substantial expression in PMN, we did not evaluate the expression of AQP1 and AQP5 in the three fractions of WBC after treatments with the inhibitors. Moreover, in the experiments of cell migration, by selectively inhibiting AQP3 and AQP9 a strong reduction in cell motility was observed, indicating that no other AQPs were compensating the blockade of AQP3 and/or AQP9.
3 - Timepoints of Bacterial Killing: The authors showed that AQP inhibition does not impair killing is important, however this could based on limited timepoints. The authors could provide data for longer-term bacterial survival (more than 1hour). I suggest to extend bacterial killing assays to longer timepoints (e.g., 2-4 hours).
Answer. We thank the reviewer for the suggestion. The current figures show time courses for up to two hours, as requested. At this point no living bacteria were left in the whole blood assay, and few were left in the PMN assay. Therefore, we did not see an indication to prolong the experiments to later timepoints.
2nd - Minor concerns
A - Figure 1: Provide the same calibration scale for the graphics on panel A, B and C. A table of the primer sequence used should be provide as supplementary material. Justify why the β-actin was used as reference instead other house-keep gene.
Answer. The same calibration scale for the graphics on panels A, B and C of Figure 1 has been provided and a Table with the primer sequences used for the RT-qPCR analyses has been provided as supplementary material. Regarding the use of β-actin as reference, a similar study by Rump et al. used b-Actin, thus allowing direct comparison of the studies.
B - Figure 2 and 3: the authors should include the positive and negative controls for primary and secondary antibody autofluorescence. The images should be provided as supplementary material.
Answer. A figure of immunofluorescence with the positive and negative controls for primary and secondary antibodies has been prepared, inserted as supplementary material (Figure 1 supplementary material) and mentioned in the main text of the manuscript.
C - Figure 8 and 9: In the panel A of the figure 8 and figure 9, the author should provide survival curve of K. pneumoniae for all the experimental groups (vehicle, 1% DMSO, and AQP3/AQP9 selective inhibitors).
Answer. The survival curves of K. pneumoniae for all the experimental groups have been provided in the panel A of figures 8 and 9.
D - Figure 10: Provide a calibration scale for the representative micrographs of the on LPS-induced cell migration assay.
Answer. A calibration scale (bars) has been added in the figure in question.
E - AQP3 vs. AQP9: It’s not clear how specific the inhibitors are for AQP3 vs. AQP9. Are off-target effects ruled out? A short discussion should be included on the relevance of AQP3/AQP9 selective inhibitors.
Answer. We have evaluated the specificity among homologous AQPs in previous studies and the related references have been added (lines 491-495).
F - Selectivy: Include data or discussion validating the selectivity of DFP00173, HTS13286, and RG100204.
Answer. References speaking to good selectivity have been added (lines 491-505). However, this aspect has currently not been investigated in depth.
G - Mechanistic Depth: The article lacks mechanistic insight into how AQP3/AQP9 affect migration. Is it through cytoskeletal rearrangement, volume regulation, or signaling modulation? Discussion regarding the potential pathways could be provide.
Answer. As mentioned for point 1, unravelling the precise molecular mechanisms through which, distinctly, AQP3 and AQP9 intervene in the migration of human leukocytes remains challenging. This task will be the object of further studies. However, at the end of the discussion (lines 549-563) we have expanded the part on the possible mechanisms on how AQP3/AQP9 could intervene and in addition to a couple of reviews on the topic we have cited a series of works that have provided useful insights on the issue.
H - Discussion: The authors could provide more insights and explore more therapeutic implications. Can AQP inhibition modulate pathological inflammation without impairing host defense?
Answer. A sentence has been added to the conclusions section. We would however like avoid overt speculation regarding the translatability of such findings.
Reviewer 2 Report
Comments and Suggestions for Authors
Dear Authors,
This is an interesting article in which the authors investigated the function of AQP9 and AQP3 in leukocytes, with a particular focus on cell migration and phagocytosis. It is not easy to clarify the functional importance of aquaporins because their function was sometimes redundant. The authors showed that both AQP3 and AQP9 inhibition caused significant impairment of phagocytosis of K. pneumoniae in monocytes. They also found that both AQP3 and AQP9 were involved in leukocyte migration. This reviewer does not believe that these results completely reflect actual in vivo events. However, results are reliable as in vitro experiments.
This reviewer would like to point out a couple of issues.
#1. Lines103: Please cite literature on HTS13286 and DFP00173.
#2. Figure2: How did the authors judge AQP9 was present on the cell membrane? AQP9 localization is not apparent in these images. There might be some AQP9 on the cell membrane, but there seems to be some bright signal on the intracellular granules.
#3. Lines 322-323: “bacterial ingestion was accompanied by distinct changes in cell size and morphology compared to neutrophils exposed to medium alone” Please describe the specifics and details of the changes in neutrophils that have ingested bacteria.
Thank you very much.
Author Response
Reviewer 2
Comments and Suggestions for Authors
Dear Authors,
This is an interesting article in which the authors investigated the function of AQP9 and AQP3 in leukocytes, with a particular focus on cell migration and phagocytosis. It is not easy to clarify the functional importance of aquaporins because their function was sometimes redundant. The authors showed that both AQP3 and AQP9 inhibition caused significant impairment of phagocytosis of K. pneumoniae in monocytes. They also found that both AQP3 and AQP9 were involved in leukocyte migration. This reviewer does not believe that these results completely reflect actual in vivo events. However, results are reliable as in vitro experiments.
This reviewer would like to point out a couple of issues.
#1. Lines103: Please cite literature on HTS13286 and DFP00173.
Answer. References speaking to good selectivity have been added (lines 491-505), as already answered to the same question asked by Reviewer 1.
#2. Figure2: How did the authors judge AQP9 was present on the cell membrane? AQP9 localization is not apparent in these images. There might be some AQP9 on the cell membrane, but there seems to be some bright signal on the intracellular granules.
Answer. We thank the Reviewer for raising this point. We agree that in the images shown in Figures 2 and 3 AQP9 is also localized in the intracellular compartment of neutrophils and in monocytes and lymphocytes. We have now clearly specified this in both the figures and the text and have also discussed it in the Discussion section (.."The AQP9 immunoreactivity seen at the intracellular level suggests that the translocation of this AQP to the plasma membrane is regulated, in line with a previous work showing that phosphorylation of AQP9 at S11 through a Rac1-dependent pathway mediates membrane localization and neutrophil polarization (Karlsson et al, JLB 2011_ref. 46) (lines 486-490)."
#3. Lines 322-323: “bacterial ingestion was accompanied by distinct changes in cell size and morphology compared to neutrophils exposed to medium alone” Please describe the specifics and details of the changes in neutrophils that have ingested bacteria.
Answer. The specifics and details of the changes in neutrophils that have ingested bacteria have been now described in the manuscript as follow: “..Thereby, bacterial ingestion was accompanied by distinct changes in cell size and morphology compared to neutrophils exposed to medium alone (Figure 5A, B). This can be explained by bacterial LPS that induce better adherence of granulocytes and thereby changes of the infected granulocytes (Dahinden and Fehr, 1983_ref. 44) (lines 330-335)
Round 2
Reviewer 1 Report
Comments and Suggestions for Authors
The new version of the manuscript followed all the previous concerns and is well accepted.